# A Reinforcement Learning-based Bidding Strategy for Data Consumers in Auction-based Federated Learning

**Xiaoli Tang[1], Han Yu[1], Xiaoxiao Li[2,3]**
[1]College of Computing and Data Science, Nanyang Technological University, Singapore
[2]Department of Electrical and Computer Engineering, University of British Columbia, Canada
[3]Vector Institute, Canada

## Abstract

Auction-based Federated Learning (AFL) fosters collaboration among self-interested data consumers (DCs) and data owners (DOs). A major challenge in AFL pertains to how DCs select and bid for DOs. Existing methods are generally static, making them ill-suited for dynamic AFL markets. To address this issue, we propose the Reinforcement Learning-based Bidding Strategy for DCs in Auction-based Federated Learning (`RLB-AFL`). We incorporate historical states into a Deep Q-Network to capture sequential information critical for bidding decisions. To mitigate state space sparsity, where specific states rarely reoccur for each DC during auctions, we incorporate the Gaussian Mixture Model into `RLB-AFL`. This facilitates soft clustering on sequential states, reducing the state space dimensionality and easing exploration and action-value function approximation. In addition, we enhance the $\epsilon$-greedy policy to help the `RLB-AFL` agent balance exploitation and exploration, enabling it to be more adaptable in the AFL decision-making process. Extensive experiments under 6 widely used benchmark datasets demonstrate that `RLB-AFL` achieves superior performance compared to 8 state-of-the-art approaches. It outperforms the best baseline by 10.56% and 3.15% in terms of average total utility.

## 1   Introduction

Driven by stringent user privacy and data confidentiality requirements, federated learning (FL) has recently attracted substantial attention from both academic and industrial domains [1–7]. With data owners (DOs), also known as FL clients, being self-interested entities that weigh a myriad of factors (ranging from costs to potential utility gains) when deciding which FL data consumer (DC) to collaborate with, the design of FL incentive mechanisms [8, 9] has taken center stage. These mechanisms aim to incentivize DOs to participate in FL through various reward strategies.

Auction-based federated learning (AFL) is an important sub-field of FL incentive mechanism design, due to its potential to achieve both efficiency and fairness [10, 11]. In AFL, the incentive mechanism between DOs and DCs is organized in the form of an auction. This process is overseen by an auctioneer, who acts as an intermediary to facilitate the exchange of data for FL training. AFL methods can be roughly categorized into three main groups [12]: 1) DC-side methods, 2) auctioneer-side methods, and 3) DO-side methods. The DC-side methods focus on how DCs select and place bids on DOs, aiming to optimize key performance indicators (KPIs) while staying within budget constraints. The auctioneer-side methods optimize DC-DO matching and pricing strategies, along with the design of effective auction mechanisms. The goal is to achieve specific operational objectives, such as maximizing social welfare or minimizing social costs, for the AFL ecosystem. DOs care more about determining the allocation of local resources and setting their reserve prices for profit maximization [13].

39th Conference on Neural Information Processing Systems (NeurIPS 2025).

In recent times, there has been a growing research interest [14, 15] in investigating DC-side issues, developing optimal bidding strategies to assist them in effectively bidding for DOs. However, existing methods are generally static approaches. They are essentially represented by either non-linear or linear functions with parameters derived from historical auction data using heuristic techniques. Then, these parameters are applied to new auctions, even if the dynamics of these new auctions might vary significantly from those in the historical data. In practice, the inherent dynamism of AFL markets poses a considerable challenge for static DC bidding methods to consistently achieve desirable outcomes.

To bridge this important gap, we propose a Reinforcement Learning-based Bidding Strategy for DCs in Auction-based FL (RLB-AFL). It incorporates historical states into a Deep Q-Network to capture sequential information critical for AFL DC bidding decisions. To mitigate the state space sparsity issue in AFL, where specific states rarely re-appear for a DC, we propose to integrate the Gaussian Mixture Model into RLB-AFL to enable soft clustering on sequential states, thereby reducing the state space dimensionality, which in turn eases exploration and action-value function approximation. Moreover, we improve the $\epsilon$-greedy policy to help an RLB-AFL agent strike a balance between exploitation and exploration, enhancing its applicability in the decision-making process for each DC within an AFL ecosystem.

To our best knowledge, RLB-AFL is the first cluster-based reinforcement learning approach that facilitates a large number of DCs to compete for a common pool of DOs. Extensive experiments conducted on 6 widely used benchmark datasets demonstrate the superiority of RLB-AFL compared to 8 state-of-the-art existing approaches. It outperforms the best baseline by 10.56% and 3.15% in terms of average total utility and model accuracy, respectively.

## 2  Related Works

Existing AFL DC-side methods, the primary focus of this paper, can be broadly categorized into two main groups based on the auction mechanism adopted: 1) those for reverse auction scenarios, and 2) those for forward auction scenarios.

Methods like [16–27] designed for reverse auction scenarios aim to help the DC select DOs after receiving their asking profiles (which may include available data resources and the corresponding asking prices). [24] combined a quality-aware model aggregation algorithm with reverse auction, and proposed the FAIR method. It employs a greedy algorithm based on Myerson's theorem [28] to determine the winning DOs and maximize the valuation for the DC. However, a crucial limitation of these methods arises from their assumption that there is only one DC and multiple DOs in the AFL marketplace. This monopoly market assumption is unrealistic in practice, where multiple DCs are typically present.

Methods like [29, 14, 30, 31, 15] focus on assisting DCs in bidding for DOs under a competitive AFL market setting, employing forward auction mechanisms. These methods design bidding strategies to guide DCs in determining bid prices for DOs. [15] introduced the Fed-Bidder bidding strategy which considers DC budget constraints, DO relevance and prior auction-related knowledge to design a bidding function. It also emphasized the critical roles played by accurate estimation of DO utility and the selection of an appropriate winning function in shaping optimal bidding strategies.

RLB-AFL falls into the category of methods designed for forward auction scenarios. However, it is noteworthy that most existing bidding strategies designed for DCs are static, and thus may not be suitable for dynamic AFL markets.

## 3  Preliminaries

**AFL Market**: Generally, an AFL market consists of three types of participants [12]: 1) Data Owners (DOs): entities possessing potentially sensitive yet valuable data, who are willing to share or sell access to their data resources for FL task training in exchange for appropriate compensation. 2) Data Consumers (DCs): organizations or individuals requiring data to train their machine learning models via FL. 3) Auctioneer: a trusted third-party entity orchestrating the auction process between DOs and DCs. It facilitates the exchange of data resources for FL training tasks through an auction mechanism, such as the Second-Price Sealed-Bid (SPSB) auction.

When a DO is ready to offer its services for FL task training, it notifies the auctioneer, specifying its bid request and the reserve price.[1] The auctioneer then announces the auction to all DCs currently participating in the AFL market. Any DC whose required the corresponding data resources aligns with the DO's offering submits a bid for the auction.

Each DO can trigger the following auction process: 1) **Bid Request Initiation**: DO $i \in [C_s]$ generates a bid request about itself (e.g., identity, data quantity, etc.) and sends it along with the the reserve price (i.e., the lowest price it is willing to accept for selling the corresponding resources [32]) to the auctioneer. 2) **Bid Request Dissemination**: The auctioneer disseminates the received bid request to the relevant DCs whose FL tasks are relevant to the data resources of the DO being auctioned. 3) **Bidding Response**: Each relevant DC evaluates the potential value and cost of the received bid request, and decides on a bid price based on its bidding strategy. The DCs submit their bids to the auctioneer. When a DC has exhausted its budget, it will forfeit future auctions. 4) **Outcome Determination**: Upon receiving bids from relevant DCs, the auctioneer determines the winning price based on an auction mechanism. It then compares the winning price with the reserve price set by each DO. If the winning price is lower than the reserve price, the auctioneer terminates the auction and informs the DO to initiate another auction for the same resources. Otherwise, the auctioneer informs the winning DC about the cost (i.e., the winning price) it needs to pay, informs the losing DCs, and informs the DO about the winning DC it shall join.

The FL training process for the target DC based on its recruited DOs commences once either their budget is depleted or all available DOs have been recruited by DCs.

**Problem Formulation**: The AFL DC bidding can be framed as an optimization problem within budget $B$ limit [15] to maximize the DC's total utility with respect to a set of DOs $[1, C]$:

$$\max \sum_{i \in [1,C]} x^i \times v^i, \quad s.t. \sum_{i \in [1,C]} x^i \times p^i \leq B, \tag{1}$$

where $v^i$ denotes the utility the DC can gain from DO $i$ being auctioned. The specific process to calculate $v^i$ is described in the subsequent section. Here, $x^i \in \{0, 1\}$ denotes whether the target DC wins $i$, and $p^i$ denotes the payment from the target DC to $i$. Notably, under the SPSB auction mechanism [33], if a DC wins the bid for a DO $i$, $p^i$ equals to the second-highest bid price among all the bids received by the auctioneer; otherwise, $p^i = 0$.

In [15], it has been shown that under SPSB, the optimal bidding strategy is:

$$b^i = v^i / \omega. \tag{2}$$

$\omega$ is a scaling factor. When the sequence of DO arrival is known in advance, the optimal $\omega$ value ($\omega^*$) can be determined using a greedy approximation algorithm [34]. Unfortunately, in practice, strategies must be executed in real-time without prior knowledge of the available data resources being auctioned. Moreover, the auction environment typically exhibits high nonstationarity due to the dynamic behaviors of all participating DCs, making the derivation of $\omega^*$ challenging.

**Data Owner Reputation Modeling**: Following [35], we assess the utility of attracting a DO $i$ for a DC by evaluating $i$'s reputation. To calculate $i$'s reputation ($\phi^i$) for a DC, we start by adopting the computationally efficient GTG-Shapley method [36], which measures $i$'s contribution to the DC in the Shapley Value sense [37]. This value is then fed into the Beta Reputation System (BRS) [38] to obtain $i$ reputation value.

The contribution of a DO $i$ to a DC can be grouped into two categories: 1) negative (i.e., $\phi^i < 0$) and 2) positive (i.e., $\phi^i \geq 0$). We adopt the variables $nc^i$ and $pc^i$ to record the number of negative contributions and the number of positive contributions made by $i$ for the DC, respectively. Then, we employ the BRS to compute the reputation value $v_i$ for DO $i$ on the target DC as:

$$v^i = \mathbb{E}[Beta(pc^i + 1, nc^i + 1)] = \frac{pc^i + 1}{pc^i + nc^i + 2}. \tag{3}$$

It is essential to emphasize that, as illustrated in Eq. (3), DO $i$'s reputation undergoes dynamic updates throughout the FL model training process. Additionally, in situations where no prior information is accessible, the default initial reputation value of $i$ is established as the uniform distribution, represented by $v^i = U(0, 1) = Beta(1, 1)$.

---

[1]Following [15], we assume that DOs arrive and make their bid requests sequentially, one after the other.

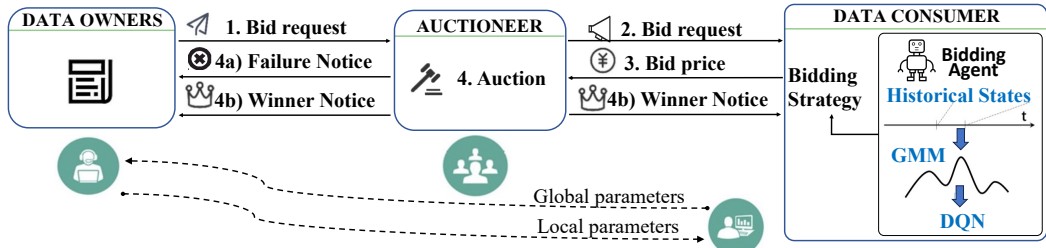

Figure 1: The `RLB-AFL` system architecture.

## 4 The Proposed `RLB-AFL` Approach

The system architecture of `RLB-AFL` is illustrated in Fig. 1. In response to the limitations of current DC bidding methods within dynamic AFL markets, as well as the challenges arising from the lack of access to private information about DOs and the underlying AFL system dynamics, we propose a reinforcement learning (RL) framework, `RLB-AFL`, for modeling each DC's bidding process as an $\omega$ control problem. RL is pivotal in tackling this problem due to its ability to learn optimal decision-making policies directly from interactions with the environment. Following [29], we leverage the Deep Q-Network (DQN) [39] as the underlying RL model, which uses deep neural networks to approximate the optimal action-value function, enabling effective decision-making in complex, high-dimensional state spaces. Our design incorporates Gaussian Mixture Model (GMM) into `RLB-AFL` to perform soft clustering on the continuous state space, mitigating the curse of dimensionality. In addition, we improve the $\epsilon$-greedy policy to strike a balance between exploration and exploitation, enabling `RLB-AFL` to adapt to new market conditions while capitalizing on its learned knowledge.

### 4.1 POMDP Modeling

`RLB-AFL` frames the bidding agent for the target DC by sequentially regulating $\omega$ under the Partially Observable Markov Decision Process (POMDP) setting [40]. The objective of the DC agent is to acquire an optimal $\omega$ controlling policy that maximizes the accumulated reward $\sum_{i=1}^{C} \gamma^{i-1} r^i$, while ensuring that $\sum_{i=1}^{C} x^i \times p^i \leq B$. The fundamental components of the POMDP are:

- $o_i / s_i{}^2$: Before the bid for DO $i$, the observation $o_i$ consists of: 1) $B^i$: the remaining budget, 2) $(C - i)$: the remaining DOs, and 3) $v^i$: the utility the target DC can gain from DO $i$, and is formulated as:

$$o_i = (B^i, C - i, v^i). \tag{4}$$

- $a_i$: We introduce several adjustment rates to $\omega$, often taking the form of an action $a \in \mathcal{A}$ as $\omega_i = \omega_{i-1} \times (1 + \lambda_a)$, where $\lambda_a$ represents the adjustment rate related to $a$.

- $r_i$: The reward for bidding for DO $i$ is computed as $r_i = x^i \times v^i$, where $x^i \in \{0, 1\}$ indicates whether the DC wins the auction.

- $\gamma$: The objective of the target DC is to maximize the overall utility of winning DOs, subject to the budget constraint $B$, irrespective of utility over time. Therefore, the reward discount factor $\gamma$ is set as 1, i.e., $\gamma = 1$.

Although the state formulation in Eq. (4) is straightforward, it resides within an infinitely continuous state space. In light of this, the occurrence of a particular state might be rare in AFL processes, particularly when sequential information is incorporated. Consequently, accurately learning an approximation of the value function becomes extremely challenging. Furthermore, this issue leads to high exploration costs. Therefore, it is crucial to map the state space into a lower-dimensional and, ideally, finite space. To tackle this issue, we propose a state clustering method.

---

²In this paper, while it might be more accurate to refer to it as an observation, we continue to use the both term state and term observation concerning the bidding process without ambiguity.

## 4.2 State Clustering

During the decision-making process, an agent can rely on historical information. This intuition has been widely adopted in multiple domains (e.g., recommendation systems [41], click-through rate estimation in computational advertising [42]), and has been proven effective. This suggests that it might be useful to base DC bidding strategies on the historical information from recent auctions, rather than only on the current state. Motivated by this observation, we frame the state from the perspective of modeling the sequential information of states (i.e., historical states). Specifically, we define the sequential information of states as:

$$\boldsymbol{s}_i = \langle s_{i-W+1}, \cdots, s_i \rangle, \tag{5}$$

where $W$ is a hyperparameter representing the window size. If $i - j < 0$ ($0 < j \leq W - 1$), $s_{i-j}$ is configured as a zero vector. Following this, we obtain the combined state $\hat{s}_i$ through the application of the state mapping function $f_{state}(\cdot)$ as:

$$\hat{s}_i = f_{state}(\boldsymbol{s}_i). \tag{6}$$

In the context of AFL, an intuitive observation is that similar historical state sequences tend to yield comparable rewards under a given bidding strategy. This key insight emphasizes the potential benefits of grouping similar state sequences together through clustering. However, within an AFL market, each DC faces a continuous state space comprising numerous elements (e.g., utility derived from auctioned data, remaining budget, available time steps). Navigating this extensive state space and learning an effective bidding strategy is a formidable challenge, as it requires capturing the intricate dynamics and stochasticity of the environment without direct access to DOs' private information.

To address this critical issue, we design a soft clustering approach over the historical state sequences based on GMM [43]. It effectively reduces the dimensionality of the state space, while preserving the essential information encoded in the sequential state trajectories. Dimensionality reduction is crucial for mitigating the state space sparsity issue, which can otherwise hinder accurate value function approximation and incur excessive exploration costs during the RL process. The ability of GMM to model complex data distributions through a mixture of Gaussian components makes it well-suited for clustering the continuous and high-dimensional state representations arising from the incorporation of historical state information.

Specifically, let $K$ denote the number of clusters of the historical states, and $\{\boldsymbol{s}_1, \boldsymbol{s}_2, \cdots, \boldsymbol{s}_N\}$ denote the available historical states. The conditional probability of each historical state $\boldsymbol{s}_i$ for a cluster $k$ is modeled by a Gaussian distribution:

$$p(\boldsymbol{s}_i | z_i = k; \mu_k, \Sigma_k) = \mathcal{N}(\mu_k, \Sigma_k). \tag{7}$$

Alternatively, the prior probability of each cluster $k$ is assumed to adhere to a Multinomial distribution $Multinomial(u)$, with:

$$u_k > 0, \quad \sum_k u_k = 1, \quad p(z_i = k) = u_k. \tag{8}$$

Then, the total log-likelihood of the historical states is:

$$L(u, \mu, \Sigma) = \sum_{i=1}^{N} \log \sum_{k=1}^{K} \mathcal{N}(\mu_k, \Sigma_k) u_k. \tag{9}$$

Following the Expectation-Maximization (EM) algorithm, `RLB-AFL` gradually learns the parameters $u$, $\mu$ and $\Sigma$. Specifically, in the E-step, the weight recording the affinity of historical state $\boldsymbol{s}_i$ to cluster $k$ is calculated as:

$$w_k^i = p(z_i = k | \boldsymbol{s}_i; u, \mu, \Sigma) = \frac{\mathcal{N}(\mu_k, \Sigma_k) u_k}{\sum_{j=1}^{K} \mathcal{N}(\mu_j, \Sigma_j) u_j}. \tag{10}$$

In the M-step, the parameters of cluster $k$ are updated as:

$$u_k = \frac{1}{N} \sum_{i=1}^{N} w_k^i, \quad \mu_k = \frac{\sum_{i=1}^{N} w_k^i \boldsymbol{s}_i}{\sum_{i=1}^{N} w_k^i}, \quad \Sigma_k = \frac{\sum_{i=1}^{N} w_k^i (\boldsymbol{s}_i - \mu_k)(\boldsymbol{s}_i - \mu_k)^i}{\sum_{i=1}^{N} w_k^i}. \tag{11}$$

The E-step and the M-step are iteratively repeated until convergence. Eventually, the weight vector $\Gamma(\boldsymbol{s}_i)$ expressing the inclination of $\boldsymbol{s}_i$ towards each cluster $k \in [1, K]$, is:

$$\Gamma(\boldsymbol{s}_i) = (w_1^i, w_2^i, \cdots, w_K^i). \tag{12}$$

However, obtaining a set of historical states and subsequently estimating the parameters of the GMM in a sequential manner is not practical for DCs newly joining the AFL process. Therefore, we also propose the following EM algorithm to dynamically update the GMM clusters in an adaptive fashion. Given the current state $\boldsymbol{s}_i$, utilizing the prevailing GMM, we begin by computing the posterior probabilities $\Gamma(\boldsymbol{s}_i)$ as outlined in Eq. (10). Then, the M-step is enhanced as (the first two equations have been consolidated into a single line):

$$u_k = (1 - \kappa)u_k + \kappa w_k^i, \quad \mu_k = \mu_k + \kappa w_k^i \boldsymbol{s}_i, \quad \Sigma_k = \Sigma_k + \kappa w_k^i (\boldsymbol{s}_i - \mu_k)(\boldsymbol{s}_i - \mu_k)^i, \tag{13}$$

where $\kappa \in \{0, 1\}$ denotes the hyperparameter balancing the weight assigned to the incoming instance.

### 4.3 Enhanced $\epsilon$-greedy Policy

DQN implements the $\epsilon$-greedy policy to strike a balance between exploitation and exploration, where the agent selects action $a^* = \arg\max_a Q(o, a)$ with a probability of $(1 - \epsilon)$, while taking a random action with a probability of $\epsilon$. The parameter $\epsilon$ is typically set to a larger value and slowly anneals over time to a smaller value. Yet, determining an appropriate annealing rate is crucial, as a high annealing rate limits exploration, while a low one can lead to slow policy convergence.

However, in the context of AFL DC bidding, the optimal bidding theory guarantees a consistent optimal $\omega^*$ for each DO $i \in [1, C]$. Given the observation $o_i$, taking the optimal action is equivalent to adjusting $\omega$ to approach $\omega^*$. Any deviation from this optimal action results in a reduction in potential value, as reflected by a lower $Q$ value. Hence, considering our action space $\mathcal{A}$, which encompasses a range of adjustment rates denoted as $\{\lambda_a\}$), the action-value distribution $Q(o_i, a_i)$ across the action space $\mathcal{A}$ sorted according to the adjustment scale $\lambda_a$ of action, should ideally exhibit unimodality [44]. Therefore, if the distribution is not unimodal, it implies an abnormal estimation of $Q$. It is necessary to increase $\epsilon$ to encourage exploration in this state.

---

**Algorithm 1** `RLB-AFL`

---

Initialize $Q(o, a; \theta)$ and its target network with $\hat{\theta} \leftarrow \theta$, update frequency of target network $\tau$, replay memory $\mathcal{D}$, training batch size $m$.

1: Initialize $\omega_0$;
2: **for** $i = 1$ to $C$ **do**
3:     Obtain the state $s_i$ based on the GMM;
4:     Compute $a_i$ according to the enhanced $\epsilon$-greedy policy w.r.t $Q$;
5:     Obtain $\omega_i$ based on $\omega_{i-1}$;
6:     Calculate $b_i$ according to Eq. (2);
7:     Submit $b^i$ to the auctioneer;
8:     Get reward $r_i$ and the payment $p^i$;
9:     Store transition tuples in $\mathcal{D}$;
10:     Sample a random minibatch of $m$ samples from $\mathcal{D}$;
11:     $y = r + \gamma \max_{a'} Q(o, a'; \hat{\theta})$;
12:     Update $\theta$ by minimizing $\sum_m [(y - Q(o, a; \theta))^2]$;
13:     $\hat{\theta} \leftarrow \theta$ every $\tau$ steps;
14: **end for**

---

### 4.4 The Training Process of `RLB-AFL`

`RLB-AFL` is based on DQN. The action-value function $Q(o, a)$ is modeled by a deep neural network (DNN) with parameter $\theta$. To enhance training stability, we leverage a target network parameterized by $\hat{\theta}$, adopting a similar DNN architecture to approximate $Q(o, a)$ (Algorithm 1).

Updating of the parameters $\theta$ is achieved by minimizing $\mathcal{L}(\theta) = \frac{1}{2}\mathbb{E}_{(o,a,r,o')\sim\mathcal{D}}[(y - Q(o, a; \theta))^2]$. Here, $\mathcal{D}$ is a replay buffer that stores transition tuples $\langle o, a, r, o' \rangle$, where $o'$ denotes the new observation of the bidding agent following the action $a$, derived from the initial observation $o$ and

corresponding reward $r$. Randomly sampling batches of transitions during training, buffer $\mathcal{D}$ facilitates learning from past experience. Let $\gamma$ denote the discount factor. The temporal difference target $y$ is computed as $y = r + \gamma \max_{a'} Q(o', a'; \hat{\theta})$. $Q(o', a'; \hat{\theta})$ denotes the predicted action-value function of the DC bidding agent for its subsequent observation $o'$ considering all feasible actions $a'$. $\hat{\theta}$ denotes the parameters of the target network, and $Q(o', a'; \hat{\theta})$ denotes the predicted action-value function. The target network ensures training stability by maintaining a fixed target throughout the training process, periodically updated to synchronize with the current action-value network.

## 5 Experimental Evaluation

### 5.1 Experiment Settings

**Datasets**: We adopt six widely used datasets in FL studies: 1) MNIST[3], 2) CIFAR-10[4], 3) Fashion-MNIST (a.k.a. FMNIST) [45], 4) EMNIST-digits (a.k.a. EMNISTD) [46], 5) EMNIST-letters (a.k.a. EMNISTL) [46] and 6) Kuzushiji-MNIST (a.k.a. KMNIST) [47]. The FL models used are the same as those employed in [15].

**Comparison Baselines**: We compare `RLB-AFL` against the following eight well-established bidding approaches: 1) Constant Bid (**Const**) [48], 2) Randomly Generated Bid (**Rand**) [22, 23], 3) Below Max Utility Bid (**Bmub**) [49], 4) Linear-Form Bid (**Lin**) [50], 5) Bidding Machine (**BM**) [51], 6) Reinforcement Learning-based Bid (**RLB**) [29, 14, 52]. More detailed descriptions of these methods can be found in [29]. In addition, we include **Fed-Bidder** [15] which is specifically designed for AFL DCs. It guides DCs to competitively bid for DOs to maximize their utility under a given budget constraint. Fed-Bidder is implemented as two variants: 7) Fed-Bidder-sim **(FBs)** with a simple winning function and 8) Fed-Bidder-com **(FBc)** with a complex winning function.

**Experimental Scenarios**: We conduct experiments under two scenarios, each involving 10,000 DOs: 1) **IID data**: Each DO possesses a set of 1,000 images, including some noisy ones. To facilitate the effective evaluation of DOs' reputations by DCs, the 10,000 DOs are organized into five groups, each comprising 2,000 DOs. In addition, different percentages of noisy data are introduced for each DO group as follows: DOs in the first, second, third, fourth, and last groups each owns 0%, 10%, 25%, 40%, and 60% noisy data, respectively. 2) **Non-IID data**: By adjusting the class distribution among individual DOs, which hold 1,000 images, we intentionally introduce data heterogeneity in this experimental setup. Following [35], the Non-IID setup is implemented as follows: one class (for datasets except EMNISTL) or six classes (for EMNISTL) are designated as the minority class, assigned to 100 DOs. Therefore, images for all classes are possessed by these 100 DOs, while the other nine or twenty classes except the minority class, are exclusively held by all other DOs. Scenarios in which the minority classes are with 10% or 25% noisy data are also included.

**Implementation Details**: To deal with the challenge of lacking a publicly available dataset related to AFL, we conducted simulations where we tracked the behaviors of DCs under the setting of forward auction and generalized second-price sealed-bid (SPSB) over time in four distinct scenarios, each involving 160 DCs. 1) One-eighth of the DCs adopts each of the eight comparison approaches. 2) Three-sixteenths of the total population adopt each of RLB, FBs, FBc and BM, while one-sixteenth of all DCs adopt the other four approaches. 3) Custom-tailored for AFL with both Fed-Bidder variants and `RLB-AFL`, we fine-tuned the ratio of DCs choosing FBc and FBs to surpass those opting for the remaining six baseline methods. Specifically, 50 DCs chose FBc and FBs each, while the other six baselines were adopted by 10 DCs each. 4) Following the settings in Scenario 3, 65 DCs chose FBc and FBs each, while the other six baselines were adopted by 5 DCs each..

To assess the efficacy of `RLB-AFL`, 9 AFL DCs are implemented, each employing one of the previously mentioned bidding methods to bid for DOs. For the action-value function utilized by the bidding agent, `RLB-AFL` employs fully connected neural networks. These networks consist of three hidden layers, each comprising 64 nodes. The RMSprop with a 0.0001 learning rate is adopted to train all the neural networks. The discount factor $\gamma$ for the reward is set to 1, as the primary aim of a DC is to maximize the overall utility gained from winning DOs within the budget constraints. A replay buffer $\mathcal{D}$ of size 6,000 is used for training the action-value function $Q$ (i.e., $|\mathcal{D}| = 6,000$). During training, 32 samples from $\mathcal{D}$ are utilized for updating $Q$ at each training step (i.e., $m = 32$). Furthermore, the

---

[3]http://yann.lecun.com/exdb/mnist/
[4]https://www.cs.toronto.edu/kriz/cifar.html

Table 1: Comparison results of the total utilities and FL model accuracy (%) across different datasets and budget settings under the IID scenario. "Bud" means budget and "Acc" means accuracy. The best results are highlighted in **Bold**.

| Bud | Method | MNIST | | CIFAR | | FMNIST | | EMNISTD | | EMNISTL | | KMNIST | |
|---|---|---|---|---|---|---|---|---|---|---|---|---|---|
| | | Utility | Acc | Utility | Acc | Utility | Acc | Utility | Acc | Utility | Acc | Utility | Acc |
| 100 | Const | 7.28 | 78.03 | 6.68 | 35.28 | 7.53 | 69.95 | 7.32 | 78.52 | 7.53 | 68.82 | 7.05 | 62.67 |
| | Rand | 6.73 | 73.23 | 7.40 | 34.93 | 9.02 | 70.55 | 7.84 | 79.83 | 8.08 | 67.40 | 8.25 | 61.52 |
| | Bmub | 8.48 | 80.72 | 9.67 | 35.74 | 9.56 | 71.31 | 9.45 | 80.36 | 9.97 | 70.39 | 9.12 | 63.54 |
| | Lin | 11.42 | 82.02 | 10.96 | 37.70 | 11.14 | 71.84 | 11.15 | 80.76 | 11.22 | 71.23 | 11.18 | 64.19 |
| | BM | 13.21 | 83.07 | 13.61 | 38.30 | 13.83 | 73.81 | 12.96 | 81.27 | 14.10 | 72.08 | 14.24 | 66.02 |
| | FBs | 15.22 | 83.12 | 14.66 | 39.78 | 15.05 | 73.82 | 14.83 | 81.65 | 14.89 | 73.19 | 14.99 | 68.91 |
| | FBc | 15.16 | 83.38 | 15.72 | 40.33 | 15.23 | 74.63 | 14.80 | 81.66 | 14.90 | 73.23 | 14.89 | 68.63 |
| | RLB | 15.91 | 83.24 | 15.30 | 40.24 | 15.36 | 74.18 | 15.41 | 81.96 | 15.33 | 73.36 | 15.71 | 68.25 |
| | RLB–AFL | **18.62** | **85.86** | **16.84** | **41.83** | **17.68** | **76.82** | **17.95** | **83.69** | **17.56** | **74.79** | **17.37** | **70.66** |
| | w/o c | 16.97 | 84.42 | 15.98 | 41.11 | 16.71 | 75.68 | 16.64 | 82.37 | 16.48 | 73.82 | 16.04 | 69.34 |
| | w/o e $\epsilon$ | 16.25 | 84.55 | 16.25 | 41.09 | 16.29 | 75.79 | 16.58 | 83.46 | 15.37 | 73.59 | 16.33 | 70.26 |
| 200 | Const | 9.04 | 81.00 | 9.18 | 37.81 | 9.34 | 69.06 | 8.03 | 79.32 | 9.22 | 70.94 | 8.65 | 63.45 |
| | Rand | 8.50 | 81.10 | 8.67 | 38.60 | 10.87 | 71.20 | 8.45 | 79.80 | 9.23 | 70.76 | 9.58 | 61.87 |
| | Bmub | 12.08 | 81.90 | 10.72 | 39.39 | 11.48 | 72.23 | 10.15 | 81.37 | 11.63 | 71.99 | 10.48 | 64.74 |
| | Lin | 13.80 | 82.13 | 13.43 | 40.13 | 13.55 | 72.85 | 13.29 | 81.40 | 13.56 | 73.05 | 13.62 | 68.99 |
| | BM | 15.64 | 84.55 | 16.02 | 41.33 | 16.55 | 75.29 | 15.45 | 82.46 | 16.96 | 73.56 | 17.22 | 71.76 |
| | FBs | 18.53 | 84.36 | 17.73 | 42.24 | 18.35 | 75.36 | 17.84 | 82.26 | 17.88 | 73.93 | 18.08 | 71.98 |
| | FBc | 18.15 | 84.53 | 17.53 | 42.12 | 18.48 | 75.25 | 17.55 | 82.10 | 17.80 | 73.79 | 17.76 | 72.16 |
| | RLB | 18.46 | 85.14 | 18.03 | 42.47 | 18.40 | 75.03 | 18.17 | 82.60 | 18.23 | 74.47 | 18.65 | 74.60 |
| | RLB–AFL | **19.98** | **86.89** | **20.72** | **44.69** | **21.31** | **77.48** | **20.37** | **84.48** | **21.77** | **77.88** | **20.86** | **75.84** |
| | w/o c | 19.18 | 85.75 | 19.46 | 43.55 | 20.93 | 76.26 | 19.25 | 84.19 | 20.41 | 76.15 | 19.81 | 74.80 |
| | w/o e $\epsilon$ | 19.24 | 86.19 | 19.58 | 43.86 | 19.96 | 76.86 | 19.75 | 83.83 | 19.35 | 75.88 | 19.92 | 75.32 |
| 400 | Const | 7.43 | 81.25 | 8.39 | 39.03 | 8.91 | 70.92 | 7.50 | 80.14 | 9.10 | 71.69 | 8.27 | 69.26 |
| | Rand | 10.76 | 80.22 | 7.08 | 39.61 | 10.47 | 71.03 | 7.48 | 79.75 | 8.11 | 72.15 | 8.79 | 71.58 |
| | Bmub | 11.56 | 82.30 | 10.33 | 40.14 | 11.35 | 73.20 | 10.51 | 82.05 | 11.88 | 73.32 | 10.70 | 72.66 |
| | Lin | 14.77 | 83.31 | 14.35 | 41.65 | 14.38 | 75.33 | 14.13 | 82.04 | 14.39 | 73.94 | 14.52 | 72.78 |
| | BM | 17.07 | 84.85 | 17.04 | 42.68 | 17.20 | 75.40 | 16.25 | 82.78 | 17.82 | 74.57 | 18.54 | 73.87 |
| | FBs | 19.58 | 85.14 | 18.66 | 43.86 | 19.28 | 76.74 | 18.73 | 83.51 | 18.73 | 75.12 | 19.05 | 74.17 |
| | FBc | 19.31 | 85.20 | 18.45 | 43.83 | 19.34 | 76.31 | 18.52 | 83.42 | 18.63 | 75.20 | 18.71 | 73.95 |
| | RLB | 19.83 | 85.77 | 18.97 | 43.70 | 19.42 | 77.10 | 19.15 | 83.70 | 19.06 | 75.06 | 19.68 | 75.94 |
| | RLB–AFL | **22.06** | **87.63** | **20.14** | **45.94** | **21.67** | **79.47** | **20.71** | **85.24** | **21.65** | **77.91** | **20.91** | **77.89** |
| | w/o c | 21.94 | 86.39 | 19.10 | 44.37 | 20.38 | 78.11 | 20.24 | 84.99 | 20.22 | 76.69 | 20.52 | 77.38 |
| | w/o e $\epsilon$ | 20.43 | 86.28 | 19.56 | 44.46 | 20.79 | 78.75 | 20.53 | 84.17 | 20.86 | 77.03 | 20.44 | 76.52 |

target network for $Q$ is updated once every 30 training steps (i.e., $C = 30$). The window size for historical states is fixed at 40 (i.e., $W = 40$), and the number of clusters $K$ is set to 10 (i.e., $K = 10$). The weight assigned to the incoming instance during the M-step is set to 0.5 (i.e., $\kappa = 0.5$). Each recruited DO undergoes 30 local training epochs, with a batch size of 256.

**Evaluation Metrics**: We employ the following two metrics to assess the compared approaches: 1) **Utility**: It quantifies the total reputation of DOs enlisted by the corresponding target DC upon reaching either the bid request limits or the budget limit. 2) **Test Accuracy (Acc)**: Acc denotes the accuracy of the FL models achieved until reaching either the budget limit or the limits on bid requests.

## 5.2 Results and Discussion

To perform a comprehensive comparison of all nine bidding methods, experiments are carried out on six datasets with budgets varying from low to high among $\{100, 200, 400\}$.

Table 1 illustrates the outcomes of various comparison methods under the IID scenario. It can be observed that the proposed RLB-AFL method consistently achieves the best performance among all the comparison methods in terms of both test accuracy and utility across all three budget settings and all six datasets. In particular, compared to the best-performing baseline, RLB-AFL improves the total utility and the test accuracy of the resulting FL model by 12.18% and 2.93%, respectively. Table 2 illustrates the outcomes of various comparison methods under the Non-IID scenario. The results align with the performance shown in Table 1 with the proposed RLB-AFL improving the test accuracy by 3.19% on average under the Non-IID scenario.

Const and Rand perform poorly compared to other methods due to their disregard for DOs' utility in their formulation. Among all the other comparison methods, Bmub and Lin exhibit inferior performance, with Lin being more effective than Bmub. This can be attributed primarily to the introduction of randomness in the bidding strategy of Bmub. The remaining five comparison methods

Table 2: FL model accuracy (%) comparison across different datasets and budget settings under the Non-IID scenario. "Bud" means budget. *10% and 25%* represent 10% and 25% noisy data, respectively.

| Bud | Method | MNIST | | CIFAR | | FMNIST | | EMNISTD | | EMNISTL | | KMNIST | |
|---|---|---|---|---|---|---|---|---|---|---|---|---|---|
| | | 10% | 25% | 10% | 25% | 10% | 25% | 10% | 25% | 10% | 25% | 10% | 25% |
| 100 | Const | 66.09 | 70.06 | 13.04 | 13.66 | 59.98 | 59.31 | 76.94 | 76.67 | 64.25 | 63.76 | 60.12 | 59.22 |
| | Rand | 68.77 | 67.10 | 10.61 | 10.76 | 61.36 | 60.77 | 75.58 | 78.24 | 63.50 | 63.15 | 59.19 | 58.52 |
| | Bmub | 70.10 | 70.85 | 15.02 | 13.63 | 62.12 | 61.60 | 77.27 | 77.76 | 66.18 | 65.68 | 63.09 | 61.78 |
| | Lin | 71.95 | 71.14 | 18.47 | 17.76 | 64.08 | 64.09 | 78.45 | 77.88 | 65.47 | 64.75 | 63.23 | 62.90 |
| | BM | 72.18 | 71.91 | 19.52 | 19.59 | 66.89 | 66.69 | 79.35 | 78.83 | 66.11 | 65.35 | 64.57 | 63.99 |
| | FBs | 73.05 | 72.68 | 23.37 | 22.48 | 70.67 | 70.54 | 79.34 | 78.78 | 67.34 | 66.63 | 65.86 | 64.75 |
| | FBc | 73.45 | 74.12 | 23.25 | 22.59 | 71.04 | 70.92 | 79.77 | 79.30 | 66.48 | 65.61 | 65.21 | 64.33 |
| | RLB | 73.78 | 73.94 | 23.57 | 22.97 | 71.41 | 71.70 | 79.86 | 79.10 | 67.10 | 66.32 | 65.98 | 64.34 |
| | RLB-AFL | **74.84** | **74.88** | **25.79** | **25.42** | **72.94** | **73.58** | **80.46** | **81.80** | **69.22** | **67.47** | **68.38** | **65.39** |
| | w/o c | 74.13 | 74.24 | 24.66 | 23.95 | 72.33 | 72.61 | 80.05 | 80.58 | 68.29 | 66.94 | 66.88 | 64.86 |
| | w/o e $\epsilon$ | 74.36 | 74.52 | 24.83 | 24.28 | 72.47 | 72.84 | 80.19 | 80.73 | 68.46 | 67.03 | 67.26 | 65.12 |
| 200 | Const | 69.86 | 68.12 | 10.74 | 10.97 | 62.25 | 61.59 | 77.91 | 77.69 | 67.14 | 66.64 | 61.33 | 58.33 |
| | Rand | 69.38 | 69.17 | 10.31 | 10.28 | 62.10 | 61.32 | 78.56 | 78.35 | 67.68 | 67.27 | 62.11 | 58.42 |
| | Bmub | 71.55 | 71.04 | 13.34 | 13.13 | 63.14 | 62.84 | 79.22 | 78.74 | 68.39 | 67.89 | 64.68 | 63.25 |
| | Lin | 72.49 | 71.52 | 18.91 | 18.28 | 64.32 | 64.27 | 79.28 | 78.80 | 69.33 | 68.88 | 67.58 | 66.37 |
| | BM | 73.24 | 72.86 | 20.33 | 20.20 | 66.81 | 67.82 | 80.36 | 79.82 | 69.00 | 68.16 | 68.39 | 68.02 |
| | FBs | 74.19 | 73.18 | 23.67 | 23.08 | 71.70 | 71.79 | 80.13 | 79.65 | 68.79 | 68.17 | 68.95 | 69.00 |
| | FBc | 74.03 | 73.51 | 23.46 | 22.87 | 71.76 | 71.71 | 80.25 | 79.84 | 69.76 | 69.09 | 68.63 | 67.53 |
| | RLB | 75.16 | 73.69 | 23.68 | 23.32 | 71.20 | 71.05 | 80.34 | 79.88 | 69.29 | 68.69 | 69.61 | 70.49 |
| | RLB-AFL | **77.62** | **75.67** | **24.88** | **25.64** | **73.93** | **73.97** | **82.86** | **81.55** | **71.41** | **70.72** | **71.75** | **71.86** |
| | w/o c | 75.93 | 74.44 | 23.92 | 24.58 | 72.02 | 72.68 | 81.49 | 80.38 | 70.52 | 69.26 | 70.46 | 70.98 |
| | w/o e $\epsilon$ | 77.34 | 74.68 | 24.41 | 24.76 | 72.55 | 72.94 | 82.23 | 80.77 | 70.93 | 69.84 | 71.11 | 71.26 |
| 400 | Const | 70.52 | 69.43 | 17.07 | 16.99 | 61.89 | 60.96 | 78.42 | 78.17 | 67.56 | 67.12 | 67.92 | 68.42 |
| | Rand | 69.59 | 68.66 | 20.84 | 20.58 | 62.72 | 62.05 | 78.59 | 78.50 | 68.36 | 67.99 | 69.34 | 69.92 |
| | Bmub | 71.87 | 71.06 | 21.96 | 20.98 | 63.86 | 63.64 | 79.82 | 79.30 | 69.04 | 68.57 | 69.25 | 68.95 |
| | Lin | 72.60 | 71.81 | 24.00 | 23.29 | 65.49 | 65.47 | 79.86 | 79.37 | 69.94 | 69.52 | 69.90 | 69.46 |
| | BM | 74.57 | 73.79 | 25.33 | 24.27 | 66.92 | 67.38 | 80.76 | 80.28 | 71.09 | 70.71 | 71.12 | 70.77 |
| | FBs | 75.39 | 74.46 | 26.19 | 25.06 | 71.03 | 70.72 | 81.19 | 80.65 | 71.09 | 70.62 | 71.27 | 71.16 |
| | FBc | 75.28 | 74.53 | 25.93 | 24.82 | 72.00 | 71.98 | 81.13 | 80.60 | 71.26 | 70.81 | 70.77 | 69.97 |
| | RLB | 75.19 | 75.08 | 26.50 | 25.39 | 72.28 | 72.27 | 81.40 | 80.88 | 71.40 | 70.99 | 71.75 | 71.21 |
| | RLB-AFL | **76.54** | **76.43** | **27.86** | **26.79** | **73.76** | **74.78** | **82.69** | **82.13** | **74.25** | **73.39** | **72.99** | **72.15** |
| | w/o c | 75.89 | 75.94 | 27.12 | 26.05 | 72.77 | 73.41 | 81.99 | 81.50 | 72.68 | 72.31 | 72.06 | 71.66 |
| | w/o e $\epsilon$ | 76.07 | 76.15 | 27.39 | 26.22 | 73.16 | 73.95 | 82.33 | 81.74 | 73.07 | 72.69 | 72.37 | 71.94 |

consistently exhibit superior performance compared to the aforementioned four simpler approaches. This improved performance can be attributed to the incorporation of auction records, which encompass both bidding records and auction history, as well as the adoption of machine learning/reinforcement learning frameworks. BM, FBs, and FBc underperform RLB and RLB-AFL as they belong to the category of static bidding methods, lacking adaptability to the highly dynamic auction environment of AFL. Compared to BM, FBs and FBc perform better. This can be attributed to the fact that these two methods use a specially designed bidding function to model the market price distribution, enhancing the accuracy of bid cost expectations. In BM, the market price distribution is obtained by marginalizing the prediction of the market price density of bid requests, which may result in overfitting. Nevertheless, these three bidding methods are formulated as either non-linear or linear functions, trained on historical auction data utilising heuristic approaches. When these functions are exposed to new auctions, which might differ from the historical ones due to the dynamism of the AFL market, achieving consistent desired outcomes becomes challenging. Although RLB adopts dynamic programming to enhance its bidding process, it is not specifically designed for AFL DCs, and might face challenges related to state sparsity, potentially leading to poor performance in AFL settings. This limitation has been effectively addressed by RLB-AFL. Furthermore, RLB-AFL integrates an enhanced $\epsilon$-greedy policy into its framework to achieve an advantageous trade-off between exploration and exploitation.

**Ablation Study**: We created two ablated versions of RLB-AFL: 1) **w/o c**: excluding the states clustering part from RLB-AFL. 2) **w/o e** $\epsilon$: the proposed $\epsilon$-greedy policy in RLB-AFL is replaced by the general $\epsilon$-greedy policy. These modifications is to examine the impact of incorporating the states clustering operation and the enhanced $\epsilon$-greedy policy into RLB-AFL. Tables 1 and 2 present the results. It can be observed that RLB-AFL outperforms its ablated variants in terms of the total utility and accuracy of FL models. Therefore, the two proposed designs are effective and improve the performance of RLB-AFL.

**Sensitivity Analysis on Number of Clusters**: To see the impact of the GMM cluster number on `RLB-AFL`, we vary the number of GMM clusters from $\{3, 5, 10, 15, 20\}$. The averaged accuracy of the FL models under the 400 budget settings is shown in Table 3. Initially, as the number of clusters increases, there is a noticeable ascent in accuracy, followed by a subsequent decline. This trend suggests that a higher cluster count initially yield a more precise state mapping function. However, excessive growth leads to increased GMM representation size and state sparsity. Within our experimental framework, it becomes apparent that selecting a cluster size ranging between 10 to 15 leads to optimal outcomes for the model user. This range strikes a balance, steering clear of the limitations associated with a smaller cluster count, while also avoiding the over-expansion of GMM representation that triggers state sparsity.

Table 3: Accuracy (%) under various number of GMM clusters ($K$).

| $K$ | MNIST | CIFAR | FMNIST | EMNISTD | EMNISTL | KMNIST |
|---|---|---|---|---|---|---|
| 3 | 85.54 | 41.62 | 75.93 | 83.11 | 73.49 | 70.33 |
| 5 | 86.09 | 43.88 | 77.42 | 83.85 | 74.84 | 72.41 |
| 10 | 87.63 | 45.94 | 79.47 | 85.24 | 77.91 | 77.89 |
| 15 | 86.21 | 43.46 | 78.38 | 82.93 | 75.72 | 75.84 |
| 20 | 85.32 | 43.02 | 76.59 | 81.60 | 74.65 | 73.92 |

# 6    Conclusions

To address the limitations of static bidding strategies in dynamic AFL markets, we propose `RLB-AFL`, a novel RL-based bidding method for DCs. It frames bidding as a $\omega$-control problem using a DQN architecture. Given the high-dimensional, continuous state space, including utility, budget, and time, training a generalizable RL model is challenging. `RLB-AFL` tackles this with Gaussian mixture model-based soft clustering and a refined $\epsilon$-greedy policy to balance exploration and exploitation. However, while `RLB-AFL` and other methods focus solely on competition among DCs, they overlook potential collaboration, which can indirectly influence behavior. Future work will incorporate these complex inter-DC relationships.

# Acknowledgments

The research is supported, in part, by the Ministry of Education, Singapore, under its Academic Research Fund Tier 1 (RG101/24); the RIE2025 Industry Alignment Fund – Industry Collaboration Projects (IAF-ICP) (Award I2301E0026), administered by A*STAR, as well as supported by Alibaba Group and NTU Singapore through Alibaba-NTU Global e-Sustainability CorpLab (ANGEL).

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
