# OpenReview forum: "A Reinforcement Learning-based Bidding Strategy for Data Consumers in Auction-based Federated Learning"
_NeurIPS.cc/2025/Conference — NeurIPS 2025 poster_

### Official Review · Reviewer_ntVa · 2025-06-08

**Clarity:** 3
**Significance:** 3
**Originality:** 3
**Rating:** 4
**Confidence:** 3

**Summary:**

This paper proposes RLB-AFL, a Reinforcement Learning-based Bidding Strategy tailored for data consumers (DCs) in auction-based Federated Learning (AFL). The method employs a Deep Q-Network (DQN) augmented with a Gaussian Mixture Model (GMM) to manage high-dimensional and sparse state spaces and introduces an enhanced ϵ-greedy policy for better exploration-exploitation tradeoff. Experiments on six datasets show that RLB-AFL significantly outperforms eight baselines under both IID and non-IID settings.

**Questions:**

1. This work describes the process as a POMDP. But in my understanding, the state in POMDP is NOT equal to the observation, as the agent cannot observe all the state information, different from Line 160. Could authors explain this point clearly?

2. How much improvement does GMM gain over other clustering methods?

**Ethical Concerns:**

["NO or VERY MINOR ethics concerns only"]

**Final Justification:**

I believe this paper makes proper contribution. But two simplifications make me maintain reservations about its robustness and practicality in complex, real-world applications.

**Limitations:**

Yes

**Quality:**

3

**Strengths And Weaknesses:**

Strength:

1. The proposed method effectively mitigates the state space sparsity issue in AFL.

2. The experiment results demonstrate the superiority of RLB-AFL compared with baselines.

Weakness:

1. The authors choose GMM as the clustering method. While effective, it relies on a good initialization point and requires high computational complexity. The GMM may not scale well as the number of clusters or the dimensionality increases.

2. The policy is learned solely based on historical data, but ignores the features and potential influence of data at the data owners in the FL process.

3. While this work considers multiple DCs, the method is designed for a single DC in nature, if I understand correctly.
In the auction, each DC's decision is made based on its own "individual perspective", without modeling the strategies of other DCs or the joint dependencies of the system state.

4. DQN may struggle to converge, especially in environments with sparse states. But the authors do not show the convergence process of the proposed method.

---

> ### Author Rebuttal · Authors · 2025-07-29
>
> We are grateful for your valuable and supportive feedback. Below, we provide detailed clarifications and responses to the issues you have raised.
>
> **Weakness 1 & Question 2:** We appreciate your concern and acknowledge that GMM can be sensitive to initialization and less scalable in high-dimensional settings. However, our implementation is carefully designed to remain efficient in the AFL context:
>
> 1) The state representations are low-dimensional (≤ 3), and clustering is applied to short historical sequences (e.g., window size = 40), keeping the input space compact.
>
> 2) We use a small, fixed number of clusters ($K$ = 10–20), as supported by sensitivity analysis (Table 3), which avoids overfitting and reduces computational overhead.
>
> 3) To improve robustness and efficiency, we adopt an online EM algorithm with warm-start initialization, which incrementally updates cluster parameters without full retraining.
>
> 4) Empirically, our method scales well in scenarios with up to 160 data consumers and across six benchmark datasets.
>
> 5) The per-step GMM update cost is $O(K)$, and the clustering component contributes minimally to overall system complexity.
>
> 6) We have added clarification in the revised implementation details.
>
> To further justify the effectiveness of our GMM-based clustering module in addressing state sparsity in dynamic AFL environments, we compare it with two widely used and conceptually different clustering methods: K-means and DBSCAN. These methods are selected because they are well-established, computationally comparable, and represent two distinct clustering paradigms (partition-based and density-based, respectively).
>
> We evaluate all methods on six benchmark datasets under the IID setting with a fixed budget of 400. The table below reports the averaged accuracy (%) of the resulting FL models across datasets:
>
>
> | Method      | MNIST | CIFAR  |FMNIST  |EMNISTD  |EMNISTL  |KMNIST  |
> | :---        |    :----:   |    :----:     |:----:     |:----:     |:----:     |:----:     |
> | Ours (GMM)  | 87.63       | 45.94   |79.47   |85.24   |77.91   |77.89   |
> | K-means	| 86.44	| 45.15	| 77.51	| 83.03	| 77.40	| 77.04|
> | DBSCAN	| 86.77	| 45.70	| 78.64	| 82.75	| 76.88	| 76.75|
>
> It can be observed that the original proposed method with the GMM clustering methods outperforms methods with K-means and DBSCAN clustering methods. It achieves superior accuracy of FL models.
>
> This superiority may stem from several key advantages of GMM:
>
> 1) Its soft clustering mechanism allows better handling of ambiguous and non-repetitive states common in AFL.
>
> 2) The online EM updates adapt effectively to dynamic auction patterns without requiring retraining.
>
> 3) Unlike DBSCAN, which struggles with temporal structure and parameter sensitivity, GMM offers robust generalization with a fixed, optimized cluster number.
>
> --------
>
> **Weakness 2:** We appreciate this important observation and would like to clarify that the features and potential influence of data at the data owners are indirectly but effectively reflected in the utility value $v^t_i$, which serves as a key component of our POMDP state representation.
>
> The detailed feature integration process is as follows:
>
> 1) Direct Feature Impact: According to our "Data Owner Reputation Modeling" section, the data quality and characteristics of data owners directly affect their reputation through the GTG-Shapley method
>
> 2) Reputation-Utility Mapping: The data owner's reputation $\phi_i$ captures their historical contribution quality and data characteristics through the Beta Reputation System (Eq. 3)
>
> 3) State Integration: These data owner features influence the POMDP state through the utility component $v^t_i$ in the observation $o^t_i = (B^t_i, C-i, v^t_i)$
>
> 4) Policy Learning: Consequently, our learned policy implicitly considers data owner features and their FL influence through this reputation-based utility estimation
>
> Advantages of this indirect approach involves:
>
> 1) Privacy Preservation: Avoids requiring direct access to private data owner information
>
> 2) Dynamic Adaptation: Reputation updates reflect changing data owner behavior and contribution quality over time in FL process
>
> 3) Comprehensive Feature Capture: The GTG-Shapley method captures complex, multi-dimensional aspects of data owner value beyond simple static features
>
> --------
>
> **Weakness 3:** Thanks a lot for your insightful comment. Our approach assumes that data consumers (DCs) act independently in a non-cooperative, competitive environment, which reflects realistic AFL market dynamics where entities are self-interesed and do not share strategic information.
>
> Although each DC makes decisions from its own perspective, multi-agent interactions are captured indirectly through the shared auction mechanism and dynamic reward feedback. In this way, strategic dependencies among DCs are reflected in the evolving market context.
>
> We agree that explicitly modeling other agents’ strategies or joint dependencies could be a valuable direction and will discuss this as future work in the revised manuscript.
>
> --------
>
> **Weakness 4:** Thank you very much for your insightful comment. Following your suggestion, we have plotted the average utility per episode over 200 rounds to illustrate the convergence behavior of the proposed method. This figure has been added to the revised manuscript.
>
> Due to the rebuttal policy, we are unable to include figures or external links in this response. However, the results clearly demonstrate that our method achieves stable and rapid convergence, even in dynamic and sparse AFL environments. This stability is largely attributed to the introduced state clustering mechanism and the enhanced ε-greedy exploration, which effectively mitigate state sparsity and improve sample efficiency during training.
>
> --------
>
> **Question 1:** We appreciate the reviewer’s attention to this detail. We agree that in a strict POMDP setting, the true state is not fully observable. In Line 160, we refer to the observation (i.e., the agent’s perceived local state) as the input to the policy, not the true environment state. To avoid confusion, we will revise the wording to clarify that the agent operates under partial observability, and the observation is used as a proxy for the underlying (unobservable) global state.

---

> > ### Comment · Reviewer_ntVa · 2025-08-03
> >
> > Thank you for your reply! Regarding Weakness 2 and 3, I still have some concerns.
> >
> > W2: The reputation indeed reflects historical contributions but is a lagging indicator. Given this, the model still fails to account for the instantaneous value of the data held by owners in the current round, which is important for a dynamic learning process like federated learning.
> >
> > W3: As the model does not explicitly model or predict the strategies of other agents, the concern remains for its effectiveness in a truly competitive market.
> >
> > These two simplifications make me maintain reservations about its robustness and practicality in complex, real-world scenarios. So my score remains the same.

---

### Official Review · Reviewer_HmSa · 2025-06-30

**Clarity:** 3
**Significance:** 3
**Originality:** 4
**Rating:** 6
**Confidence:** 5

**Summary:**

The authors propose a reinforcement learning (RL) based bidding method for Data Consumers (DCs), where the bidding process is modeled as an $\omega$  control problem, with $\omega$ being a scaling factor for the utility of the Data Owners (DOs). The authors employ Deep Q-learning as the underlying RL method and incorporate historical information into the state space. To enhance exploration, the authors propose a modified $\epsilon$-greedy strategy. In addition, they employ the Gaussian Mixture Model to manage state space dimensionality and facilitate the learning process. Extensive experiments demonstrate that RLB-AFL outperforms several state-of-the-art methods in terms of utility and accuracy.

**Questions:**

Q1. Do each of the DCs have access to all DOs?

Q2. What is $\phi^i$ before Eq. (3)?

Q3. In $\sum_{i=1}^C \gamma^{t-1}r^i$, is there a summation over time missing?

Q4. Can this method also be applied to other auction scenarios, such as repeated auctions?

**Ethical Concerns:**

["NO or VERY MINOR ethics concerns only"]

**Final Justification:**

he authors has addressed most of my previous concerns. Combing with other reviewers' comments, I decide to raise my score.

**Limitations:**

See weaknesses.

**Quality:**

3

**Strengths And Weaknesses:**

Strengths:

S1. The proposed dynamic AFL learning market problem is significant and addresses a meaningful challenge.The paper presents an innovative application of reinforcement learning in the context of AFL, a relatively unexplored area.

S2. The proposed RLB-AFL model demonstrates superior performance on several benchmarks, suggesting high-quality research and implementation.

S3. The paper is well-written with clear explanations of the methodologies and results.

S4. The approach could significantly impact bidding strategies in dynamic markets, especially for data-driven environments like federated learning.

Weaknesses:

W1. The application scenarios of the proposed method is not so clear.

W2. The experiments could be enhanced by including some auction-based datasets or data from AFL market.

W3. What bidding strategies did the other bidders adopt during the experiments? Did they use similar strategies or different ones?

---

> ### Author Rebuttal · Authors · 2025-07-29
>
> Thanks a lot for your encouraging and insightful feedback. Below, we provide point-by-point explanations to key questions raised in the review comments.
>
> **W1:** Indeed, the proposed method can be adapted for use in most FL settings that include an incentive mechanism with data consumers competing to bid for data owners [Ref1, Ref2, Ref3], particularly in auction-based FL scenarios. A practical real-world application is in gas usage estimation within the power generation and delivery industry [Ref4].
>
>
> **W2:** Since publicly available data from the AFL market is not accessible, and existing auction-based datasets, such as those from online advertising, are not directly applicable to a federated learning setting, we have followed established methods [Ref1, Ref2] to collect and generate the datasets used in our experiments. We will also explore additional auction-based datasets that could potentially be adapted for a federated learning setting.
>
>
> **W3:** As detailed in the experiment section, we compared diverse advanced bidding strategies such as FBs, FBc, among others, by various bidders. This setting is designed to address concerns regarding the potential impact of competitors with advanced bidding strategies. However, if all bidders adopt the same bidding strategy, the differences in their individual historical data could lead to variations in their utility estimation and winning price models. This variance might result in differences in their bidding behaviors. Moreover, in practice, different bidders target diverse data owners due to their distinct training tasks. These factors influence their bids through specific parameters and model approaches in their bidding strategies. Nevertheless, it remains an intriguing area for future research to analyze the dynamics and potential equilibria that arise when all data consumers adopt the same strategy.
>
>
> **Q1:** No, each DC only observes and interacts with the DOs that participate in the auction rounds in which the DC is active. DCs do not have global access to all DOs or their data. This reflects realistic AFL settings, where DOs arrive sequentially and bid responses are only visible to participating parties through the auction process.
>
> **Q2:** $\phi^i$ denotes the reputation score of DO $i$ as perceived by the target DC.
>
>
> **Q3:** We apologize for the confusion. In this context, $C$ represents time. As explained in the preliminaries, each DO arrival triggers one auction round. Therefore, in our POMDP formulation, each time step corresponds to a DO, and the time horizon is implicitly represented by the number of DOs, i.e., $C$.
>
>
> **Q4:** Yes, our framework is well-suited to repeated auction settings, particularly those involving sequential decision-making under uncertainty. The proposed reinforcement learning-based bidding strategy and adaptive state clustering mechanisms are designed to generalize across auction rounds, making them applicable to other repeated or dynamic auction environments beyond AFL. We will include a brief discussion of this in the revised version to highlight the method’s broader applicability.
>
> -----------
>
> [Ref1] Yutao Jiao et al. Toward an automated auction framework for wireless federated learning services market. TMC, 2020.
>
> [Ref2] Rongfei Zeng et al. Fmore: An incentive scheme of multi-dimensional auction for federated learning in MEC. In ICDCS, pages 278288, 2020.
>
> [Ref3] Palash Roy et al. Distributed task allocation in mobile device cloud exploiting federated learning and subjective logic. Journal of Systems Architecture, 113(2):doi:10.1016/j.sysarc.2020.101972, 2021.
>
> [Ref4] Hao Sun et al, "HiFi-Gas: Hierarchical Federated Learning Incentive Mechanism Enhanced Gas Usage Estimation," in Proceedings of the 36th Annual Conference on Innovative Applications of Artificial Intelligence (IAAI-24), pp. 22824-22832, 2024.

---

> > ### Comment · Reviewer_HmSa · 2025-08-03
> > **good rebuttal**
> >
> > The authors has addressed most of my previous concerns.
> > Combing with other reviewers' comments, I decide to raise my score to 6.

---

### Official Review · Reviewer_3okG · 2025-07-02

**Clarity:** 2
**Significance:** 3
**Originality:** 3
**Rating:** 4
**Confidence:** 3

**Summary:**

This paper focuses on incentive mechanisms in federated learning, particularly auction-based approaches. The authors propose a reinforcement learning-based bidding strategy, and the experimental results demonstrate the effectiveness of the proposed method.

**Questions:**

Please see weaknesses.

**Ethical Concerns:**

["NO or VERY MINOR ethics concerns only"]

**Final Justification:**

Most of my concerns have been addressed, and after reading the other reviewers' comments, I would like to raise my score to "borderline accept".

**Limitations:**

Yes.

**Paper Formatting Concerns:**

N/A.

**Quality:**

2

**Strengths And Weaknesses:**

Strengths
1. The paper is well-organized and easy to follow.

2. The authors conduct experiments on six widely used datasets to evaluate the effectiveness of the proposed method.


Weaknesses
1. The technical contribution of the paper appears to be somewhat limited. The proposed method primarily integrates several well-established techniques, namely, DQN, GMM, and the ϵ-greedy policy.

2. The authors point out a limitation of existing works, stating that they are generally static approaches (line 39), and claim that their proposed RLB-AFL approach addresses this issue. However, reinforcement learning-based bidding strategies have already been explored in prior work (e.g., [9,46]). It would be helpful for the authors to clarify the relationship between their approach and these earlier methods, highlighting what is novel about RLB-AFL.

3. Some important related works are missing from the literature review. The authors are encouraged to consider incorporating the following references into the related work section:
(1) Juan Li, Zishang Chen, Tianzi Zang, Tong Liu, Jie Wu, Yanmin Zhu: Reinforcement Learning-Based Dual-Identity Double Auction in Personalized Federated Learning. IEEE Trans. Mob. Comput. 24(5): 4086-4103 (2025)
(2) Xiaoli Tang, Han Yu: Reputation-aware Revenue Allocation for Auction-based Federated Learning. AAAI 2025: 20832-20840
(3) Xiaoli Tang, Han Yu: Fairness-Aware Reverse Auction-Based Federated Learning. IEEE Internet Things J. 12(7): 8862-8872 (2025)

4. The authors should check the formatting of references [8]–[10], as these entries currently lack author information.

---

> ### Author Rebuttal · Authors · 2025-07-29
>
> We sincerely appreciate your encouraging and insightful feedback. Please find our detailed point-by-point responses to your questions below.
>
> **Weakness 1:** We appreciate your insightful comment. While our method incorporated established tools like DQN and GMM, the novelty lies in how they are strategically incorporated and adapted to address key, previously unresolved challenges in auction-based federated learning (AFL).
>
> 1）*A novel problem formulation: $\omega$-Controlled Bidding Strategy*
>
> Instead of directly learning bid values as in prior RL-based approaches [9][46], we adopt a theoretically grounded formulation that learns a scaling factor $\omega$, which modulates the bid according to the formula $b_i = v_i / \omega$. This strategy decouples Q-values from raw utility values, leading to significantly more stable learning, especially in dynamic markets with varying utility scales. The approach is anchored in auction theory, ensuring convergence to optimal bidding strategies and robust generalization across tasks. Even without enhanced exploration (as shown in the “w/o enhanced $\epsilon$” ablation), the $\omega$-control mechanism alone outperforms all baseline methods (including the bidding method based on reinforcement learning, i,e,. RLB baseline), demonstrating its standalone effectiveness.
>
> 2）*First Solution to State Sparsity in AFL: State Clustering*
>
> We introduce GMM-based state clustering to tackle the problem of state sparsity in AFL environments, where exact states rarely repeat due to dynamic budgets, utilities, and DO arrival sequences. Our solution applies an online, adaptive GMM over historical state trajectories, enabling the agent to generalize across similar but non-identical states. This soft clustering framework improves sample efficiency, reduces redundant learning, and facilitates faster policy convergence. Empirical results confirm its value: the “w/o cluster” ablation shows consistent performance degradation, and sensitivity analysis (Table 3) supports the effectiveness and efficiency of our clustering strategy.
>
> 3）*Domain-Specific Exploration Strategy: Enhanced $\epsilon$-Greedy Policy*
>
> We propose an enhanced $\epsilon$-greedy exploration strategy tailored specifically to the unimodal nature of Q-values under $\omega$-control, a structure derived from auction theory. This domain-aware design promotes targeted exploration, especially when Q-value distributions deviate from unimodality, helping the agent escape suboptimal bidding regions. As demonstrated in the ablation study, disabling this mechanism results in clear drops in performance, confirming its contribution to both convergence speed and utility maximization.
>
> These innovations work synergistically: $\omega$-control stabilizes learning by removing value scale dependence, GMM clustering enhances generalization under sparse states, and enhanced $\epsilon$-greedy improves exploration efficiency in dynamic environments. Together, they lead to significant performance gains (on average, +10.56% in utility and +3.15% in accuracy across six benchmark datasets) demonstrating the robustness and scalability of our approach.
>
> -----------
>
> **Weakness 2:** Thank you for the insightful comment. While [9] and [46] also adopt reinforcement learning for dynamic bidding, our approach differs in the following key aspects:
>
> 1) We learn a scaling factor $\omega$ rather than directly predicting bids, which improves learning stability across varying utility scales.
>
> 2) We apply GMM-based state clustering to address state sparsity and enable generalization across similar auction contexts.
>
> 3) We design a domain-specific $\epsilon$-greedy policy that leverages the unimodal Q-value structure unique to AFL bidding.
>
> We will revise the paper to clearly articulate these differences in the Related Work section.
>
> -----------
>
> **Weakness 3:** Thank you very much for your insightful suggestions. In the Related Work section, we focus on methods designed to support data consumers (DCs) in AFL, that is, methods that help DCs make bidding or participation decisions.
>
> The first reference you provided focuses more on assisting the coordinator in decision-making, similar to auctioneer-side mechanisms from an ecosystem perspective, as classified in [Ref1]. The second reference falls clearly under auctioneer-side strategies, rather than DC-oriented methods, again based on [Ref1]. The third reference you mentioned is already included in our manuscript as [22].
>
> We sincerely appreciate your helpful input and will revise the Related Work section to clarify this classification and reasonably incorporate the first reference. We will also ensure our review covers the most recent and relevant state-of-the-art methods.
>
> -----------
>
> **Weakness 4:** Thank you for pointing this out. We apologize for the oversight. We will correct the formatting  and consistent citation styles in the revised manuscript.
>
> We appreciate your question and hope our response provides sufficient clarity.
>
> -----------
>
> [Ref1] Xiaoli Tang et al. Intelligent Agents for Auction-based Federated Learning: A Survey. IJCAI'24

---

> > ### Comment · Reviewer_3okG · 2025-08-03
> >
> > Thank you for the authors' reply. Most of my concerns have been addressed, and after reading the other reviewers' comments, I would like to raise my score to "borderline accept".

---

### Public Comment · ~Osman_Ozaltin1 · 2026-01-19
**Code Release**

Where is the code released? Could you please post the link for the code used to generate the experimental results? In the checklist, you stated: "We will release the node once the paper is accepted." to answer the open access to data and code question

---

### Decision · Program_Chairs · 2025-09-17

**Decision:**

Accept (poster)

**Comment:**

This paper proposes RLB-AFL, a reinforcement learning–based bidding strategy for auction-based federated learning, addressing the limitations of static approaches. The key contributions include a theoretically grounded control-based bidding formulation, Gaussian mixture clustering to mitigate state sparsity, and a domain-specific exploration strategy. Strengths lie in the methodological innovations, strong experimental validation across six datasets, and clear explanations. Weaknesses include limited discussion of multi-agent interactions and the absence of real AFL market data. However, the rebuttal clearly addressed reviewer concerns, clarifying novelty and adding experimental justification. The paper can be accepted, given its technical contributions, strong empirical results, and clear contributions.